# Functional Fruit Trees from the Atlantic and Amazon Forests: Selection of Potential Chestnut Trees Rich in Antioxidants, Nutrients, and Fatty Acids

**DOI:** 10.3390/foods12244422

**Published:** 2023-12-09

**Authors:** Caroline Palacio de Araujo, Ingridh Medeiros Simões, Thuanny Lins Monteiro Rosa, Tamyris de Mello, Guilherme Bravim Canal, Adésio Ferreira, João Paulo Bestete de Oliveira, Edilson Romais Schmildt, José Carlos Lopes, Tércio da Silva de Souza, Wagner Campos Otoni, Patrícia Fontes Pinheiro, Fábio Junior Moreira Novaes, Fabricio Gomes Gonçalves, Alexandre Rosa dos Santos, Rodrigo Sobreira Alexandre

**Affiliations:** 1Center of Agricultural Sciences and Engineering, Federal University of Espírito Santo/UFES, Alto Universitário, s/n, Alegre 29500-000, ES, Brazil; 2Federal Institute of Espírito Santo, Campus Alegre, BR 482, Km 47, Rive District, Alegre 29500-000, ES, Brazil; 3North University Center of Espírito Santo, Federal University of Espírito Santo/UFES, Rodovia Governador Mário Covas, Km 60, São Mateus 29932-540, ES, Brazil; 4Federal University of Viçosa/UFV, Av. Peter Henry Rolfs, s/n, Viçosa 36570-900, MG, Brazil

**Keywords:** Lecythidaceae, non-conventional edible chestnut, selenium, functional food, bioactive composition, nutrition

## Abstract

The Amazon rainforest and the biodiversity hotspot of the Atlantic Forest are home to fruit trees that produce functional foods, which are still underutilized. The present study aimed to select potential functional nut donor trees from two Brazilian chestnuts, by evaluating the nutritional and antioxidant composition of the nuts and the fatty acid profile of the oil. The nutritional characteristics, antioxidants, oil fatty acid profile, and X-ray densitometry of the nuts were evaluated, as well as the characterization of leaf and soil nutrients for each parent tree. The nut oil was evaluated through Brix (%), mass (g), yield (%), and the fatty acid profile. For *L. pisonis*, the most nutritious nuts were produced by *L. pisonis* tree 4 (N > P > K > Mg > Ca > Zn > Fe) and *L. pisonis* tree 6 (P > Ca > Mg > Mn > Zn > Cu > Fe), and for the species *L. lanceolata*, *L. lanceolata* tree 6 (N > P > Ca > Mg > Zn > Fe > Cu) and *L. lanceolata* tree 2 (P > K > Mg > Zn > Cu). In *L. pisonis*, the highest production of anthocyanins, DPPH, total phenolics, and flavonoids was obtained from the nuts of *L. pisonis* tree 4 as well as for *L. lanceolata*, from *L. lanceolata* tree 1, except for flavonoids. The Brix of the oil from the nuts of both species showed no difference between the trees and the fatty acid profile with a similar amount between saturated (48–65%) and unsaturated (34–57%) fatty acids. Both species have nuts rich in nutrients and antioxidant compounds and can be considered unconventional functional foods. The data collected in the present study confirm that the nuts of these species can replace other foods as a source of selenium.

## 1. Introduction

Brazil is a country rich in biodiversity, where the Amazon Forest and the Atlantic Forest biodiversity hotspot are located, home to native tree species that produce edible fruits [1,2]. In these forest territories, the species *Lecythis pisonis* Cambess and *Lecythis lanceolata* Poir can be found, which produce nuts with functional characteristics. The *L. pisonis* nuts are rich in essential nutrients, antioxidant compounds, and unsaturated fatty acids [3,4,5,6]. The consumption of foods with functional characteristics has great worldwide acceptance, and is responsible for numerous benefits to human health [7]. Studies on native forests that have nuts with food potential are still scarce, compared to the number of resources available in Brazilian biodiversity [6]. Both studied species have great potential to be inserted into the nut market; however, their fruits and co-products are underused. Nut extraction is a common activity in the Amazon biome, becoming a source of subsistence for countless families and also contributing to reforestation and the preservation of forest remnants [6].

The nutritional and functional characterization of unexplored nuts allows the insertion of new products into the human diet and allows for their balanced consumption [8]. Nuts are natural antioxidants and their daily intake in correct amounts helps in the prevention of cardiovascular and inflammatory diseases, besides having antimicrobial activity [9,10]. Cancer, diabetes, and chronic inflammation are related to people’s lifestyles and an unbalanced diet. As a result, the food industry seeks daily to add products with functional characteristics to the shelves, whether fresh or processed, such as fresh or toasted nuts and their derivatives, like flour, biscuits, and milk [11,12,13]. Therefore, this food is gaining prominence due to its characteristics that are beneficial to human health, as it is a rich source of bioactive compounds and essential nutrients. For instance, nuts contribute to the reduction of the risk of chronic and anti-inflammatory diseases, also acting in the reduction of their symptoms [14]. Chestnut shells (*Castanea sativa*), previously underutilized, are being studied with a view to their inclusion in diets as a nutraceutical compound, which can act by combating oxidative events in the lungs, kidneys, and blood, increasing the expression of antioxidant enzymes and decreasing lipid peroxidation caused by reactive species of oxygen. This compound also decreased metabolic disturbances by combating the increase in glucose and blood lipids [15]. The consumption of nuts is also beneficial for reducing total cholesterol, acting on the levels of low- and high-density lipoproteins [6].

Currently, a pillar of “fundamental nutrition” has been proposed, which associates general human health with the intake of foods (nuts, fruits, and vegetables) that have a synergistic nutritional matrix, which improves the absorption and action of nutrients absorbed by the intestine [16]. This can be observed in the synergistic action between vitamin C (ascorbic acid) and vitamin E (α-tocopherol) in human serum, which have a joint action against oxidative damage [17]. The importance of nutrients is also observed in bone formation, which requires daily supplementation of calcium, magnesium, phosphorus, potassium, fluoride, and vitamin D [18]. The problem of malnutrition affects people worldwide (approximately one in three people) according to the World Health Organization [19], and eating foods rich in minerals is a way to mitigate these effects.

The chemical composition of extracted chestnuts can be influenced by the location of the parent trees. This occurs because the concentrations of trace elements in the soil are variable and are influenced by soil pH, which is observed for the mineral selenium (Se), which is found in soil at safe concentrations of <5 mg kg^−1^, but slightly abundant in acidic soils [20,21]. Nuts are rich sources of Se, which acts in the production of selenoproteins, associated with maintaining the integrity of cell membranes, developing sperm cells, regulating thyroid hormones, and mitigating stress and intestinal inflammation [22]. Its deficiency is associated with some diseases that affect heart muscle tissue, bone, and colorectal cancer [22].

Many foods are also rich sources of fatty acids, which are formed by a carboxyl group (polar) attached to a hydrocarbon chain (nonpolar). They can be classified into saturated fatty acids (absence of double bond), unsaturated and polyunsaturated, with one or more than one double bonds, respectively [23]. Excessive intake of saturated fatty acids can contribute to the onset of obesity and diabetes, in addition to increasing neurological risks with the onset of dementia, while the intake of unsaturated fats reduces these risks [24]. Nuts are foods rich in unsaturated fatty acids, which are structural components of cell membrane phospholipids [25], and also, when free in the body, are used in energy production through the synthesis of adenosine triphosphate (ATP) in mitochondria [26]. The human body cannot synthesize essential fatty acids, requiring their consumption from food sources. Linoleic and alpha-linoleic acids, both polyunsaturated, are essential fatty acids found in nuts, acting, along with their derivatives (eicosapentaenoic and docosahexaenoic acids), against cardiovascular diseases such as hypertension [27].

In view of all these features, the present study aimed to select *L. pisonis* and *L. lanceolata* trees that produce nuts with functional and nutritional characteristics that are suitable for human consumption, through nutritional characterization and antioxidant and oil fatty acid profiles.

## 2. Materials and Methods

### 2.1. Location of L. pisonis and L. lanceolata Trees

Six trees of *L. pisonis* were studied, with tree 1 (19°47′5.49″ S and 41°9′16.53″ W) and tree 2 (19°47′1.045″ S and 41°9′18.411″ W) located in the Earth Orange municipality. Tree 3 (20°45′10.195″ S and 41°17′25.782″ W), tree 4 (20°41′16.82″ S and 41°20′28.616″ W), and tree 5 (20°41′29.895″ S and 41°22′6.972″ O) were located in the municipality of Cachoeiro de Itapemirim and tree 6 (20°47′27.442″ S and 41°29′27.187″ W) in the municipality of Alegre—all belonging to the state of Espírito Santo, Brazil. Six trees of *L. lanceolata*, located in the municipality of Mimoso do Sul, Espírito Santo, Brazil, were also studied. The geographic coordinates are: tree 1 (20°54′14.458″ S and 41°30′37.221″ W); tree 2 (20°54′11.909″ S and 41°30′36.662″ W); tree 3 (20°54′12.151″ S and 41°30′36.338″ W); tree 4 (20°54′10.938″ S and 41°30′35.767″ W); tree 5 (20°54′10.598″ S and 41°30′35.306″ W); and tree 6 (20°54′10.169″ S and 41°30′35.199″ W) (Figure 1).

### 2.2. Extraction of Minerals from Chestnuts and Leaves

N, P, K, Ca, Mg, S, Fe, Zn, Cu, and Mn content was measured in the chestnuts and leaves of *L. pisonis* and *L. lanceolata* [28], whereas Se was quantified only in chestnuts [29]. Briefly, 0.5 g of samples (tegument or endosperm) was digested with 10 mL nitro-perchloric solution (800 mL of nitric acid + 200 mL of perchloric acid, 4:1 ratio) in a digester block, the temperature of which was gradually increased to 200 °C. The digestion product was diluted in distilled water to 25 mL to obtain the concentrated extract. To determine P content, 0.25 mL concentrated extract was mixed with 21.25 mL distilled water, 2.5 mL solution 725 (1 g bismuth subcarbonate, 20 g ammonium molybdate, 136 mL of H_2_SO_4_ 97% distilled water to a final volume of 1000 mL), and 1 mL 2% ascorbic acid. Absorbance at 725 nm was read on a spectrophotometer (Bel Photonics 2000 UV). An analytical curve (Ŷ = 0.5131x + 0.0059, R^2^ = 0.9999) was used to perform absorbance readings.

To determine S content, 2 mL concentrated extract was mixed with 3 mL distilled water, 2.5 mL working buffer solution, and 2.5 mL working reagent. Absorbance at 420 nm was read as above. The working buffer solution was prepared by diluting 200 mL stock buffer solution (80 g MgCl_2_‧7H_2_O, 7.7 g ammonium acetate, 1.68 g KNO_3_, 56 mL 95% ethyl alcohol, distilled water to 1000 mL) to a final volume of 1000 mL. The working reagent was prepared by diluting 26.6 g barium chloride and 0.266 g gum Arabic in 200 mL distilled water.

K content was determined directly from the concentrated extract using a flame photometer (Digimed^®^ DM-61, São Paulo, Brazil). For Ca and Mg analysis, 0.25 mL concentrated extract was mixed with 22.25 mL distilled water and 2.5 mL SrCl_2_ (16,000 mg L^−1^), followed by reading on an atomic absorption spectrophotometer (Shimadzu^®^ AA-6200 CE, Kyoto, Japan). The latter was used also to read Cu, Mn, Fe, and Zn directly from the concentrated extract, whereas Se was read in a fluorimeter.

To determine N content in fresh nuts and leaves, 0.1 g from the sample was digested in 1 mL H_2_SO_4_ (sulfuric digestion) at 300 °C for 35 min, followed by the addition of 1 mL H_2_O_2_ for the second digestion at 200 °C for 15 min. Then, 1 mL concentrated extract was diluted with 19 mL distilled water, 2.5 mL sodium tartrate (10% m v^−1^), and 2.5 Nessler’s reagent. Absorbance at 480 nm was read on the spectrophotometer.

### 2.3. Extraction of Minerals from the Soil

Three soil samples were collected from each tree location at 0–20 and 20–40 cm from the soil. The material was homogenized, dried, and sieved (10 mesh = 2 mm) before subsequent chemical analysis [30]. The pH (Gehaka^®^ PG 2000, São Paulo, Brazil) was measured in water on 10 cm^3^ of air-dried fine soil stirred in 25 mL distilled water and left to rest for 1 h before reading in a pot. To determine C content in the soil, 0.5 g of crushed and sieved soil was weighed at 80-mesh (0.177 mm), mixed with K_2_Cr_2_O_7_ (0.2 M), and left to rest on a hot plate for 10 min. After cooling, 80 mL distilled water, 1 mL H_3_PO_4_, and 3 drops of diphenylamine indicator were added and titrated in (NH_4_)_2_Fe(SO_4_)_2_‧6H_2_O (0.05 M). The blank was measured by excluding the sample.

To determine exchangeable Ca, Mg, and Al, 10 cm^3^ soil was mixed with KCl (1 M), followed by agitation and rest for 16 h. Then, 0.5 mL extract was mixed with 10 mL SrCl_2_ to determine Ca and Mg content using the atomic absorption spectrophotometer. To quantify Al, 25 mL extract was mixed with 3 drops bromothymol blue indicator (1% m v^−1^) and titrated in NaOH (0.025 M). For the determination of H + Al, 5 cm^3^ soil was mixed with 75 mL calcium acetate (0.5 M, pH 7.0), followed by agitation and 16 h of rest. Subsequently, 2 drops phenolphthalein were added to 25 mL extract and titrated with NaOH as above.

For the extraction of P, Na, K, Fe, Cu, Zn, and Mn, 10 cm^3^ soil was mixed with 100 mL Mehlich^−1^ extracting solution (0.05 M HCl, 0.0125 M H_2_SO_4_), agitated, and left to rest for 16 h. Na and K were read on the flame photometer. To determine P content, 5 mL extract was mixed with 5 mL working reagent (200 mL solution 725, 1.6 g ascorbic acid, distilled water to 1000 mL), and absorbance at 725 nm was read on the spectrophotometer. Fe, Cu, Zn, and Mn were analyzed directly from the extract using the atomic absorption spectrophotometer.

### 2.4. Biochemical Analyses

Biochemical analyses were carried out on tegument samples crushed in a knife mill (Tecnal^®^ Wyllie Micro-TE650, São Paulo, Brazil) using material that passed through a 60-mesh sieve, and on endosperm crushed in a blender at 18,000 rpm. Total phenolic content was determined according to the Folin–Ciocalteu method [31]. Briefly, a 0.25 g sample was mixed with 5 mL HCl (2 M) and kept in a water bath at 95 °C for 30 min, followed by cooling and filtering. Then, 50 μL filtrate (extract) and 2.5 mL Folin–Ciocalteu reagent (1:10 v v^−1^) were pipetted into test tubes. After 5 min, 2 mL Na_2_CO_3_ (4% m v^−1^) was added and the solution was incubated for 2 h in the dark. Total phenolic content was quantified in triplicate at 760 nm using a UV-VIS spectrophotometer (EDUTEC^®^ EEQ-9023, Brazil). An analytical curve (Ŷ = 0.04193 + 0.76973x; R = 0.99) was prepared with gallic acid (0.1–1 mg mL^−1^) and the content was expressed in an mg gallic acid equivalents 100 g^−1^ sample.

For the analysis of anthocyanins and flavonols, a 0.150 g sample was mixed with 4 mL extracting solution (96% ethanol in 1.5 M HCl, 85:15). The samples were homogenized, left to rest for 24 h in the dark, and filtered before analysis in the UV-VIS spectrophotometer at 535 nm (anthocyanin) and 374 nm (flavonols). Anthocyanin content was determined according to Equation (1), where A = anthocyanin, abs = absorbance, V = volume of the extracting solution (in mL), L = optical path (1 cm), and m = mass of the sample (in g) [32].
(1)$$A\left(mg\;100\;g^{-1}\right)=\frac{\left(abs*V*105\right)}{982*L*m},$$
whereas flavonoid (FL) content was determined using the Equation (2):(2)$$FL\left(mg\;100\;g^{-1}\right)=\frac{\left(abs*V*105\right)}{766*L*m}.$$

To determine condensed tannin (CT) content, 1 g of sample was mixed with 30 mL extracting solution (methanol:water 8:2), kept in the dark for 24 h, filtered, and placed in a water bath at 40 °C for 60 min to evaporate the methanol. A 1 mL aliquot was removed from the crude extract and diluted in distilled water to a final volume of 25 mL. Triplicates of the extract (2 mL) were diluted with 4 mL vanillin solution (10 g L^−1^) in 70% H_2_SO_4_ (v v^−1^) and placed in a water bath at 50 °C for 15 min. Absorbance at 500 nm was read using the UV-VIS spectrophotometer and condensed tannin content was expressed by Equation (3). A calibration curve was prepared with catechin (2, 5, 10, 15, 20, 25, and 30 μg mL^−1^) as indicated by [33].
(3)CTmg g−1=(abs + 0.069170.01346)25∗m.

The antioxidant capacity of the tegument and endosperm of chestnuts was analyzed using the 2,2-diphenyl-1-picrylhydrazyl (DPPH) free radical capture method [34]. An initial ethanolic extract was prepared by diluting a 0.187 g sample to a final volume of 10 mL, homogenizing for 10 min, filtering, and using 1 mL of the filtrate for a second 1:10 dilution.

A control solution consisted of a blank (3 mL ethanol) and three replicates (1 mL ethanol + 2 mL 150 μM DPPH). Tegument and endosperm samples, from each tree of the species *L. pisonis* and *L. lanceolata,* consisted of a blank (1 mL filtrate + 2 mL ethanol) and three replicates (1 mL filtrate + 2 mL 150 μM DPPH). The tubes were stored in the dark for 1 h and read at 515 nm on the UV-VIS spectrophotometer. Sequestration of the DPPH free radical (%) was obtained by Equation (4):(4)$$DPPH(\%)=\frac{\left({abs}_{Control}-{abs}_{Sample}\right)}{{abs}_{Control}}*100.$$

### 2.5. X-ray Densitometry

X-ray densitometry is a non-destructive analysis, which aims to evaluate the internal morphology of seeds by investigating their density. Based on a previous study [35], the seeds were placed between two wooden samples measuring 5 × 5 × 0.5 cm, in sample holders with a capacity for six samples, in the internal compartment of the X-ray microdensitometer (GreCon^®^, DAX 6000, Alfeld, Germany). Density values were determined every 20 μm. The X-ray densitometer used has a voltage of 33 kV, current between 0 and 1 mA, a radiation angle of 11°, and initial and final collimation of beams of 100 and 50 μm, respectively. In the end, the average density was determined.

### 2.6. Extraction of the Oil

To extract the oil from each three tree of *L. pisonis* and *L. lanceolata*, four replications were used, consisting of two nuts each. The extraction was carried out cold, using a hydraulic press (Bovenau, SH, Germany). Oil Brix (%) was evaluated using a portable digital refractometer (Megabrix^®^, REDI-P, Brazil), with the initial (INM, g) and final (FNM, g) nuts mass on an analytical scale of 0.0001 g precision, aiming to obtain the oil mass extracted (OM, g) and the oil yield (OY, %) by Equation (5):(5)$$OY\left(\%\right)=\frac{\left(OM\right)}{INM}*100.$$

### 2.7. Transesterification Reaction

Fatty acid methyl esters (FAME) were prepared according to the International Olive Council [36]. Briefly, 0.1 g of oil was dissolved in 2 mL of hexane, to which a methanolic potassium hydroxide (0.2 mL, 2 M) was added. The mixture was shaken for 30 min. After this time, it was rested for 10 min and the organic phase was collected in a 2 mL vial for chromatographic analysis.

### 2.8. FAME Analysis

An Agilent 7820A gas chromatograph (GC) (Palo Alto, CA, USA) equipped with a G4513A autosampler and coupled with an Agilent 5975 network quadrupole mass spectrometer (MS) was used. Helium (99.9992% purity) carrier gas was used at 1.5 mL min^−1^ in constant flow mode. The injection used a split mode split (ratio of 1:20) at 220 °C and a 1.0 μL injection volume. A DB-Wax capillary column (polyethylene glycol (PEG) stationary phase of 30 m length, 0.25 mm i.d., and a film thickness of 0.25 μm, J&W Scientific, Agilent Technologies (Palo Alto, CA, USA) was employed with an oven program from 50 °C (hold 1 min) to 150 °C at 10 °C min^−1^, then from 150 to 240 °C (at 3 °C min^−1^), then held at 240 °C for 5 min. The MS detector and transfer line (17.2 cm) was held at 250 °C, the ion source at 230 °C, the quadrupole at 150 °C, and an ionization voltage of 70 eV was used. Mass spectra were obtained in scan mode (50−550 Da). Chromatographic data were recorded using the Agilent MassHunter Qualitative analysis B.07.00 software. The identities of the FAME composition were characterized by comparison with the NIST/EPA/NIH Mass Spectral Library databases (2.2 version, 2014) [37].

### 2.9. Statistical Analysis

This study followed a completely randomized design, and the means were analyzed using the Scott–Knott cluster test (*p* ≤ 0.05). Box plots and correlation analyses were applied to characterize the minerals and antioxidants in nuts. Differences between pairs of characters were assessed with the *t*-test at 0.1%, 1%, and 5% significance. For multivariate analyses, data were first standardized to prevent the influence of magnitude between the different variables. Subsequently, principal component analysis (PCA) was performed on the nutrients in leaves, chestnuts, and soils, generating biplots. A Tukey test was performed (*p* < 0.005) for the characteristics of nuts and oil mass, Brix, and oil yield.

## 3. Results

### 3.1. Extraction of Minerals from Leaves, Soil and Nuts

After the absorption of nutrients from the soil, they are transported to the leaves, seeds, and fruits, contributing to the production of nuts rich in macronutrients, micronutrients, and antioxidant compounds. The trees of *L. pisonis* and *L. lanceolata*, located in different regions, showed varying concentrations of nutrients in their leaves. Among the *L. pisonis* trees, *L. pisonis* tree 1 stood out for presenting higher values for the minerals P (1.58 µg g^−1^), Ca (33.75 µg g^−1^), Mg (2.91 µg g^−1^), and Mn (377.28 µg g^−1^), followed *L. pisonis* tree 5 for K (7.99 µg g^−1^), Fe (150.46 µg g^−1^), and Zn (11.77 µg g^−1^), and *L. pisonis* tree 6 for S (3.33 µg g^−1^), Cu (13.48 µg g^−1^) and Zn (9.48 µg g^−1^). In *L. lanceolata*, the highest values were observed in *L. lanceolata* tree 5 for P (2.80 µg g^−1^), K (6.43 µg g^−1^), Cu (21.14 µg g^−1^), Zn (27.44 µg g^−1^), and Fe (77.64 µg g^−1^), and *L. lanceolata* trees 2 and 3 for Mg (1.96 and 1.88 µg g^−1^, respectively), Fe (79.66 and 82.11 µg g^−1^, respectively), and Mn (139.33 and 134.96 µg g^−1^, respectively) (Figure 2). All information is also presented numerically in Appendix A.

The soil of the *L. pisonis* trees yielded the following values at 0–20 cm: P (1.01–33.96 mg dm^−3^), K (48–130 mg dm^−3^), Ca (0.29–7.89 cmol_c_ dm^−3^), Mg (0.32–1.80 cmol_c_ dm^−3^), Fe (13.12–133.21 mg dm^−3^), Zn (0.33–8.80 mg dm^−3^), Mn (6.41–81.87 mg dm^−3^), Cu (0.14–2.25 mg dm^−3^), and Mo (6.27–29.03 g kg^−1^). Again, similar values were found also at 20–40 cm: P (0.78–33.38 mg dm^−3^), K (23–209 mg dm^−3^), Ca (0.13–6.28 cmol_c_ dm^−3^), Mg (0.22–1.61 cmol_c_ dm^−3^), Fe (13.36–147.74 mg dm^−3^), Zn (0.20–5.42 mg dm^−3^), Mn (6.45–77.02 mg dm^−3^), Cu (0.15–1.55 mg dm^−3^), and Mo (3.43–25.19 g kg^−1^) (Appendix A).

The variation of nutrients present in the soil at depths 0–20 cm and 20–40 cm was similar for the trees of both species. The soil of *L. lanceolata* trees at a depth of 0–20 cm yielded the following values: P (0.74–1.80 mg dm^−3^), K (11–23 mg dm^−3^), Ca (0.04–0.10 cmol_c_ dm^−3^), Mg (0.06–0.11 cmol_c_ dm^−3^), Fe (78.89–141.98 mg dm^−3^), Zn (0.14–0.51 mg dm^−3^), Mn (2.62–5.19 mg dm^−3^), Cu (0.01–0.14 mg dm^−3^), and Mo (22.35–33.64 g kg^−1^). Similar values were found at a depth of 20–40 cm: P (0.86–1.96 mg dm^−3^), K (10–22 mg dm^−3^), Ca (0.02–0.13 cmol_c_ dm^−3^), Mg (0.06–0.12 cmol_c_ dm^−3^), Fe (66.08–120.34 mg dm^−3^), Zn (0.10–1.52 mg dm^−3^), Mn (2.27–5.70 mg dm^−3^), Cu (0.10–0.38 mg dm^−3^), and Mo (23.23–30.69 g kg^−1^) (Appendix A).

For *L. pisonis*, the highest concentrations of minerals were observed in nuts from *L. pisonis* tree 4 (N > P > K > Mg > Ca > Zn > Fe) and *L. pisonis* tree 6 (P > Ca > Mg > Mn > Zn > Cu > Fe). *L. pisonis* tree 5 obtained the highest concentration of the antioxidant Se and the minerals P > Mg > Ca. For *L. lanceolata*, the highest mineral concentrations were observed in *L. lanceolata* tree 6 (N > P > Ca > Mg > Zn > Fe > Cu) and *L. lanceolata* tree 2 (P > K > Mg > Zn > Cu). *L. lanceolata* tree 5 obtained the highest values of the antioxidant Se and the mineral Mg (Appendix A).

### 3.2. PCA Analysis

In *L. pisonis*, the PCA of the minerals present in nuts and leaves revealed that the first two components (PC1 + PC2) could explain 68.5% of the total variation. The greatest contribution was from MgC, MnF, CaC, KF, KC, PC, ZnF, ZnC, CuC, and SeC, where “C” denotes chestnuts and “F” leaves. *L. pisonis* trees 4 and 6 are associated closely with most minerals; whereas *L. pisonis* tree 5 was close to Se, which shows a strong association with Zn and K in the leaf (Figure 3a). The PCA analysis of the minerals present in chestnuts and soil at depth p1 (0–20 cm) in *L. pisonis* revealed that the first two components explained 77.2% of the total variation. The variables that contributed most to Pp1 were MgC, CaC, CTCTp1, Kp1, SBp1, Mnp1, Cap1, H + Alp1, Mgp1, Vp1, pHp1, CuC, PC, KC, MnC, and Znp1. Most chemical characteristics of the soil are associated closely with *L. pisonis* tree 5, while Se is strongly associated with CTCTp1, Pp1, and Kp1 (Figure 3b).

The PCA of the minerals present in chestnuts and soil at depth p2 (20–40 cm) in *L. pisonis* revealed that 77.4% of the total variation could be explained by the first two components. The variables that contributed most were H + Alp2, CTCTp2, CaC, MgC, Vp2, SBp2, pHp2, Mnp2, CuC, Cp2, Mop2, MnC, Pp2, Cap2, PC, Znp2, Mgp2, Alp2, mp2, KC, and Cup2. Most of them are localized to the upper right side of the PCA plot, which is close to *L. pisonis* trees 4 and 5. SeC displayed a high positive association with Kp2, CTCtp2, Cap2, and SBp2 (Figure 3c).

In *L. lanceolata*, the first two PCA components explained 71.4% of the total variation, with the strongest contribution coming from SeC, PC, KF, MnF, CuC, MgF, MgC, NC, FeC, CaF, KC, and CaC. Most nutrients present in the leaves and nuts are associated closely with *L. lanceolata* trees 1, 2, and 4. There was also a strong association between Se in the chestnut, and Zn and S in the leaves (Figure 3d). In *L. lanceolata*, the first two components explained 75.8% of total variation, with the major contributions coming from Pp1, CuC, Alp1, SBp1, MgC, FeC, Cp1, Mop1, CTCTp1, H + Alp1, Mnp1, KC, pHp1, ISNap1, SeC, PC, mp1, and Kp1. Most variables were located on the right side of the PCA plot, close to *L. lanceolata* trees 1–3; these were negatively associated with SeC, which instead was closer to *L. lanceolata* trees 1 and 4 (Figure 3e).

In *L. lanceolata*, the first two components explained 71.2% of total variation, with the main contribution coming from SBp2, FeC, MgC, Pp2, CTCtp2, Vp2, MnC, Alp2, ISNap2, Cap2, Fep2, CTCTp2, PC, SeC, H + Alp2, Znp2, Cp2, Mop2, and pHp2. Most soil variables were close to *L. lanceolata* tree 6 and had a negative association with ISNap2. SeC was close to *L. lanceolata* tree 5 and positively associated with Znp2, Cup2, and pHp2 in the soil (Figure 3f). The high values of the first two components with respect to total variation (Figure 3) strongly support the observed behavior of the trees toward the analyzed variables.

### 3.3. Minerals and Antioxidants in Chestnuts

In *L. pisonis*, tree 4 obtained the highest accumulation of antioxidants in nuts, reaching high values for the characteristics of anthocyanin, DPPH, total phenolics, and flavonoids, its tegument being rich in tannin. *L. pisonis* tree 5 obtained the tegument with the highest DPPH, flavonoids, and tannin. In *L. lanceolata*, tree 1 obtained the highest accumulation of anthocyanin, DPPH, and total phenolics in its nuts. *L. lanceolata* tree 3 had the highest DPPH and flavonoids and *L. lanceolata* tree 6 had the highest accumulation of anthocyanins, total phenolics, and tannin. In both species, DPPH was lower in the endosperm than in the tegument (Appendix A).

In *L. pisonis*, the DPPH of the endosperm (DPPHe) of chestnuts correlated positively with total phenolics of the endosperm (TPe) (r = 0.82). High positive correlations (r ≥ 0.87) were observed also between P and K (r = 0.99), Ca (r = 0.87), Mg (r = 0.90), and flavonols in the endosperm (FLe) (r = 0.90); between K and Ca (r = 0.83), Mg (r = 0.89), and FLe (r = 0.88); between Mg and Ca (r = 0.83) and FLe (r = 0.95); between Zn and Fe (r = 0.94); between total tegument phenolics (TPt) and tegument flavonols (FLt) (r = 0.93); between N and total phenolics in the endosperm (r = 0.93); and between Se and Cu (r = −0.89) (Figure 4a). In *L. lanceolata*, total tegument phenolics exhibited a high positive correlation with anthocyanins in the tegument (At) (r = 0.93), FLt (r = 0.95), and condensed tegument tannins (CTt) (r = 0.89); while DPPH of the tegument (DPPHt) correlated positively with CTt (r = 0.88). Other positive correlations included N with Zn (r = 0.85); Mg with P (r = 0.96), K (r = 0.84), Cu (r = 0.82), and Mn (r = 0.84); K with Cu (r = 0.90) and FLt (r = 0.83); and Se with P (r = −0.93), Mg (r = −0.94), and Cu (r = −0.89) (Figure 4b).

### 3.4. X-ray Densitometry in Chestnuts

The X-ray densitometry method used was efficient in the density analysis of *Lecythis* nuts (Figure 5), and different nut formats could also be observed. Density ranged from 501.29 to 655.01 kg m^−3^ in the *L. pisonis* trees (Figure 5a–f), and from 455.62 to 681.71 kg m^−3^ in the *L. lanceolata* trees (Figure 5g–l).

### 3.5. Brix and Fatty Acid Composition in Chestnut Oil

The nuts of *L. pisonis* showed a higher initial mass for *L. pisonis* tree 5 (8.79 g), which was not statistically different from *L. pisonis* trees 2 and 3 (8.75 and 7.89, respectively), and the final mass followed the same behavior. The oil mass presented the highest average in *L. pisonis* tree 5 and there was no statistical difference for the Brix between the trees, with the highest oil yield being observed for *L. pisonis* trees 4 and 6 (45.26 and 45.53%). For the species *L. lanceolata*, trees with the highest weight of nuts reached the highest oil mass. No statistical difference was observed for Brix between the trees studied and the oil yield ranged from 26.44% (*L. pisonis* tree 3) to 43.31% (Figure 6f–j).

Ten different fatty acids were identified in the chestnut *Lecythis pisonis* and *Lecythis lanceolata* oils and their percentage values were determined (Table 1). Stearic (31.8–52.4%) and oleic (24.2–50.5%) acids were predominant, followed by myristic (8–13%), heptadec-10-enoic (3–13%), vaccenic (1–2%), and other (lower than 1%) acids.

## 4. Discussion

According to the data obtained in this study, it was observed that nuts of the species *L. pisonis* and *L. lanceolata* are rich sources of minerals and antioxidant compounds. Within each species, some trees stood out in terms of mineral accumulation in *L. pisonis* (*L. pisonis* trees 4 and 6) and *L. lanceolata* (*L. lanceolata* trees 2 and 6). The chemical characteristics, size, and number of seeds vary according to the producing parent plant and its location [5,38]. The absorption of nutrients and their transport to fruits and seeds leads to foods rich in mineral compounds with potential benefits for humans [39]. Nutrient levels vary during the reproductive period. As observed by [40], the formation of *Punica granatum* L. fruits was accompanied by an increase in Ca, Mg, Mn, and Cu, which was followed by a decrease in N, P, K, S, Fe, and Zn in the leaves. In bean cultivars, Fe, Zn, and Se were reduced in the pod during maturation, indicating that they were translocated to the seed [41].

Foods rich in bioactive compounds, which complement the human diet, contribute to the proper functioning of bodily processes. N is involved in the synthesis of proteins and amino acids (glutamine, proline, and arginine) in seeds [42], and is widely found in sapucaia nuts (19.86 g 100 g^−1^) [43], Brazil nuts (15.52 g 100 g^−1^), cashew nuts (18.4 g 100 g^−1^), peanuts (25.2 g 100 g^−1^), pistachios (15.2 g 100 g^−1^), almonds (20.8 g 100 g^−1^), nuts (14.1 g 100 g^−1^), and macadamia nuts (8.7 g 100 g^−1^) [44]. The recommended daily concentrations are P (700 mg), Ca (1000 mg), Mg (400 mg), K (3500 mg), Fe (10 mg for men and 15 mg for women), and Zn (15 mg for men and 12 mg for women) [45,46].

P is involved in maintaining pH, producing reducing power in the form of inorganic phosphate, is a structural component of cell membrane phospholipids and nucleotides, participates in phosphorylation and enzyme activation processes, and its deficiency can cause anemia, infections, and muscle weakness [47]. Lysosomal Ca^2+^ mediates autophagy and any defects in this process are associated with aberrant protein formation and disease [48]. Mg is involved in the development of skeletal tissue and the transport of neurons and muscle cells [47]. K regulates osmotic pressure, cardiovascular, respiratory, digestive, endocrine, and cell division processes [47]. Zn has a neuroprotective function as it binds to radical-scavenging proteins, and its deficiency is associated with degenerative diseases such as multiple sclerosis [49].

Selenium is one of the main antioxidants present in *L. pisonis* and *L. lanceolata* nuts. Its recommended consumption is 55 µg day^−1^ in adults, with a maximum of 400 µg day^−1^ [45]. Multiplied by the average mass of the endosperm the amount of Se chestnut^−1^ was calculated as *L. pisonis* trees 1 (66.83 µg), 2 (134.35 µg), 3 (221.26 µg), 4 (191.68 µg), 5 (649.05 µg), and 6 (98.33 µg); *L. lanceolata* trees 1 (190.99 µg), 2 (21.14 µg), 3 (73.82 µg), 4 (169.84 µg), 5 (883.16 µg), and 6 (172.21 µg). For all *L. pisonis* matrices studied, consumption should be one nut per day; only *L. pisonis* tree 5 should be limited to 1/2 a chestnut day. Seemingly, the recommended daily dose of all *L. lanceolata* trees is one chestnut; however, *L. lanceolata* tree 5 should be limited to 1/3 of a chestnut, while *L. lanceolata* tree 2 can be increased to three. In studies with four *L. pisonis* trees located in the states of Piau and Minas Gerais, [3,4] reported Se concentrations of 26.4–46.9 μg g^−1^, which are well below those found in the present study. In samples from the state of Pará, [50] reported Se concentrations of 1.2–151 μg g^−1^; whereas for 71 Brazil nut trees distributed in the regions of Acre, Mato Grosso, Roraima, Amapá, and Amazonas, the levels of Se ranged from <0.5 to 146.6 μg g^−1^. The concentration of Se in the nut can be influenced by soil pH, geology, and hydrology [21,22]. Soil pH is strictly related to the Se absorption capacity by plants, as observed by [21] when studying different soil types in the Amazon Forest, noting that even in soils with lower Se concentrations, the available fraction (Se_soluble_ + Se_adsorbed_) was higher in higher pH sites (~5.5 to 6).

Nuts of *L. pisonis* tree 4 exhibited the highest values of anthocyanins, total phenolics, flavonols, and DPPH, as did nuts of *L. lanceolata* tree 4, except for flavonols. Anthocyanins are flavonoid pigments providing color to plants, fruits, seeds, and flowers. They help scavenge free radicals and are abundant in blueberries and cranberries, whose ingestion is associated with cognitive improvements [51] and reduced triglycerides, blood pressure, and glycemia in the elderly [52]. The highest total phenolics content reported by [3,4] in *L. pisonis* was 4134 mg 100 g^−1^ for the endosperm and 37,764 mg 100 g^−1^ for the tegument. These values are higher than those reported here (513.89 and 3815.53 mg 100 g^−1^, respectively, for *L. pisonis* and 403.88 and 2695.43 mg 100 g^−1^, respectively, for *L. lanceolata*). The present values for the endosperm are more similar to those of Brazil nuts (169.2 mg 100 g^−1^), cashew nuts (316.4 mg 100 g^−1^), hazelnuts (314.8 mg 100 g^−1^), macadamia nuts (497.8 mg 100 g^−1^), peanuts (645.9 mg 100 g^−1^), pine nuts (152.9 mg 100 g^−1^), and pistachios (571.8 mg 100 g^−1^); while the tegument is akin to the values for nuts (1580.5 mg 100 g^−1^) and pecans (1463.9 mg 100 g^−1^) [53].

The following flavonoid contents have been reported for Brazil nuts (107.8 mg 100 g^−1^), cashew nuts (63.7 mg 100 g^−1^), hazelnuts (113.7 mg 100 g^−1^), macadamia nuts (137.9 mg 100 g^−1^), peanuts (189.8 mg 100 g^−1^), pecans (704.7 mg 100 g^−1^), pine nuts (45.00 mg 100 g^−1^), pistachios (143.3 mg 100 g^−1^), and nuts (744.8 mg 100 g^−1^) [53]. Catechin was the most commonly found flavonoid in the seed coat of *L*. *pisonis* [3,4]. It acts as a strong antioxidant by capturing ROS and may be a chelator of metal ions [54]. Tannin content in the tegument of *L. pisonis* (mean = 5.52 mg g^−1^) and *L. lanceolata* (mean = 6.42 mg g^−1^) nuts was lower than that reported by [3,4] in the chestnut coat of *L. pisonis* (123.81 mg g^−1^). These authors did not detect this compound in the endosperm; here, only 0.49 mg g^−1^ of tannin was found in both species. Tannins are secondary metabolites that help plants defend themselves against predators, as confirmed by rodents preferring seeds with low tannin content [55]. They also exert antioxidant effects, as seen in barley seeds, whose cultivars with higher levels of condensed tannins and total phenolics showed greater antioxidant power and protection against DNA damage [56]. In the present study, tannin concentrations were higher in the tegument than in the endosperm of chestnuts, suggesting an antioxidant and protective role for seeds in their natural environment.

The antioxidant power of the *L. pisonis* chestnut skin was associated with the presence of phenolic compounds [4]. However, this protective structure is also rich in tannins, which are responsible for the characteristic brownish color of the nuts in both species. In *L. lanceolata*, DPPHt displayed an elevated positive correlation (r = 0.88) with CTt, but a highly negative one (r = −0.99) with CTe. In studies with Brazil nuts, [57] found the phenolic compounds to be more concentrated in the brown film covering the chestnut (1236.07 mg 100 g^−1^) than in the whole nut (519.11 mg 100 g^−1^). In *L. pisonis*, phenolics of the endosperm content correlated positively with N in the endosperm. Higher N content promotes the production of proteins in seeds [58], which can become more soluble and change amino acid composition in the presence of phenolic compounds [59]. A strong positive association was detected also between Fe and Zn in *L. pisonis* nuts. Studies with cashew nuts observed that these nutrients were associated with proteins and that only 1/3 was absorbed during digestion in humans [60].

In the present study, Se in chestnuts displayed a negative association with Cu in the soil at a depth of 0–20 cm, but a positive one at a depth of 20–40 cm. Se content in chestnuts can be influenced by soil pH, as well as geological and hydrological conditions. Moreover, Se correlates positively with clay and soil S at depths of 0–20 and 20–40 [20,22]. In *Brassica napus* L., pH, S, and sulfate in the soil were related to the absorption of Se, with S and sulfate lowering soil pH [61]. Plants with elevated capacity for Se accumulation possess transporters with greater affinity for Se than S [62]. The sulfur metabolic pathway is often employed for the uptake of Se as selenate, the main form of Se in soil, is absorbed by sulfate transporters, as selenite import depends on phosphate and aquaporin transporters. Hyperaccumulating species, which can contain more than 1000 µg g^−1^ of Se in their tissues, employ complex mechanisms for the absorption and metabolism of Se [63,64]. Such species accumulate seven times more Se than *L. pisonis* tree 5 (149.55 µg g^−1^) and three times more than *L. lanceolata* tree 5 (333.27 µg g^−1^).

The determination of the nuts’ density using the X-ray densitometry method was efficient. X-ray densitometry is little explored in seed analysis; however, it is a promising non-destructive technique for quantifying the density and evaluating the internal morphology of the seed, being able to identify empty and partially empty seeds and defects in the endosperm [65]. In addition to not causing damage to the seeds, the technique is fast and does not require preparation or treatment of the seed nuts for the analysis.

The fatty acid composition of *Lecythis pisonis* and *Lecythis lanceolata* oils is different from that found in the nut *Lecythis pisonis* Cambess (LPC) [3]. The main constituents are oleic (40–45%), linoleic (32–47%), and palmitic (11–15%) acids [3]. In comparison, nuts do not have the same fatty acids, such as total saturated fatty acids (47.7–65.3% *L. pisonis* and *L. lanceolata* × 43.1–58.8% LPC), monounsaturated (34.4–51.8% *L. pisonis* and *L. lanceolata* × 40.8–56.6% LPC), polyunsaturated (0.30–0.50% *L. pisonis* and *L. lanceolata* × 0.10–0.30% LPC), and the ratio between saturated and unsaturated (S/U: 0.91–1.88% *L. pisonis* and *L. lanceolata* × 0.76–1.43% LPC) values. This is mainly due to the representative presence of linoleic acid in LPC and that it is almost absent in *L. pisonis* and *L. lanceolata*, which have a wide concentration of their constituents and specific fatty acids (oleic and heptadec-10-enoic acid), and the ratio between them (myristic/palmitic, for example). This may facilitate the chemical distinction of this species from other nuts. Within the same family (Lecythidaceae), *B. excelsa* nuts present about 80% of unsaturated fatty acids, with higher concentrations of palmitic acid (14–16%), stearic (9–11%), oleic acid (28–36%) and linoleic (36–40%) [66,67,68]. Oleic and linoleic acid are extremely important in the diet, as they may be associated with a reduction in the incidence of cardiovascular diseases [69].

Nuts can be used as a source of raw material for the production of processed foods, gaining prominence in the production of pasta and bread, due to the absence of gluten in their constitution [70]. This food can be included in the diet of people with celiac disease, which is an autoimmune condition caused by inflammation of the small intestine mucosa and atrophy of its villi [71]. This group of people cannot ingest gluten, a compound derived from wheat, rye, and barley, and replacing gluten with nut flour makes these foods suitable for inclusion in their diet [71]. The addition of gluten-free flour was also used in the production of biscuits, where the addition of chestnut flour had positive effects on organoleptic characteristics and shelf life by increasing its stability [72]. Some nuts, like the *Castanea sativa* species, are rich in condensed tannins and, consequently, have high levels of polyphenols, which can be used in the treatment of human gastritis due to their anti-inflammatory potential [73].

## 5. Conclusions

Both studied species presented trees that stood out in terms of the concentration of nutrients and antioxidants present in their nuts. *L. pisonis* trees 4 and 6 and *L. lanceolata* trees 2 and 6 have the most nutritious nuts. *L. pisonis* tree 4 also showed the highest accumulation of antioxidant compounds in its nuts. Principal component analyses (PCA) showed the behavior of nutrient absorption in nuts in terms of foliar and soil minerals. *L. pisonis* trees (1, 2, 3, 4, and 6) and *L. lanceolata* trees (1, 2, 4, and 6) have adequate Se concentrations for human consumption (one walnut day-1), while the consumption of nuts from *L. pisonis* tree 5 and *L. lanceolata* tree 5 should be limited to 1/2 and 1/3 of a nut day-1, respectively. The data collected in the present study confirm that the nuts of these species are rich sources of Se and can be used in the human diet, replacing other foods. X-ray densitometry can be a useful tool for analyzing nut quality, as it allows for the measurement of the density of samples. The Brix of the nut oil of both species showed no difference between the studied trees, showing that the variation of soluble solids is low. The predominant fatty acids were stearic acid and oleic acid, followed by myristic acid, eptadic-10-enoic, and vaccenic acid. Based on the results obtained, we conclude that these two species produce nuts with beneficial nutritional and functional properties, which can be included in human consumption. The sustainable exploitation of this material is still limited and little-known, and new complementary studies are needed to insert these nuts into the market.

## Figures and Tables

**Figure 1 foods-12-04422-f001:**
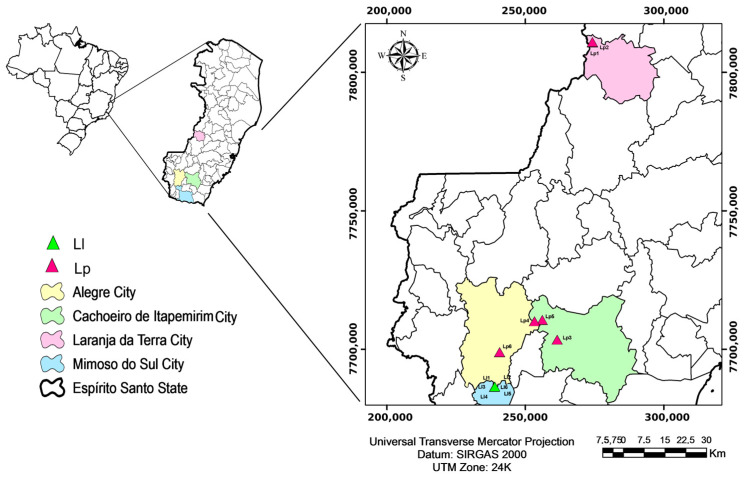
Geographical coordinates of the *L. pisonis* (Lp) and *L. lanceolata* (Ll) trees located in the state of Espírito Santo, Brazil.

**Figure 2 foods-12-04422-f002:**
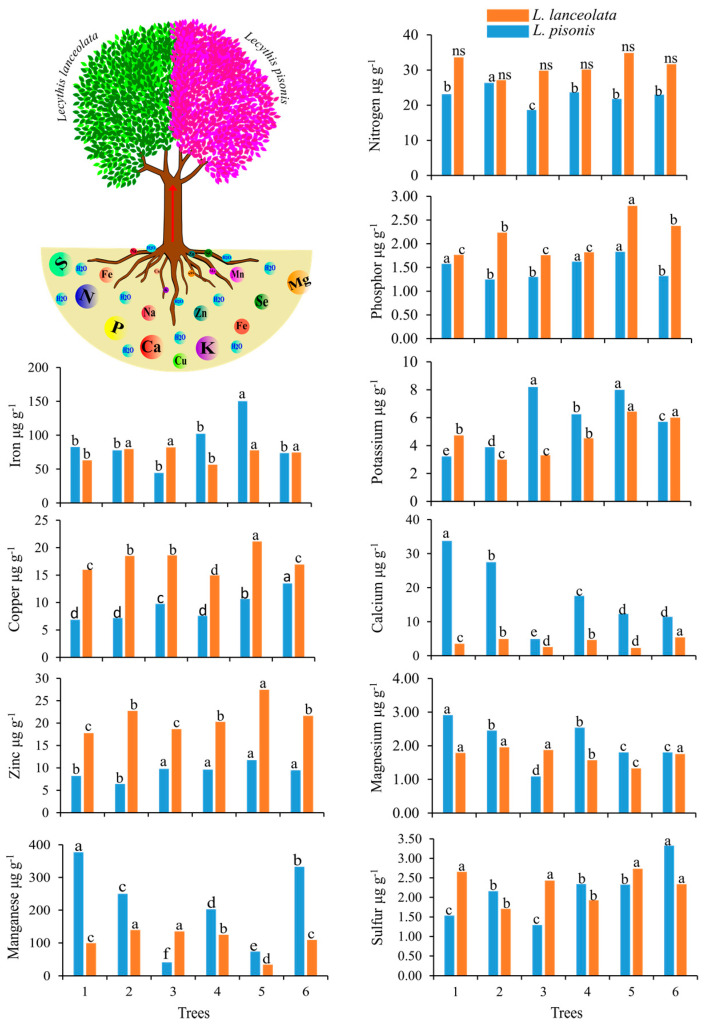
Nutrients present in the leaves of *L. pisonis* and *L. lanceolata* trees. Means followed by the same letter in the column belong to the same group of averages based on the Scott–Knott group of averages test (*p* ≤ 0.05). ns = not significant.

**Figure 3 foods-12-04422-f003:**
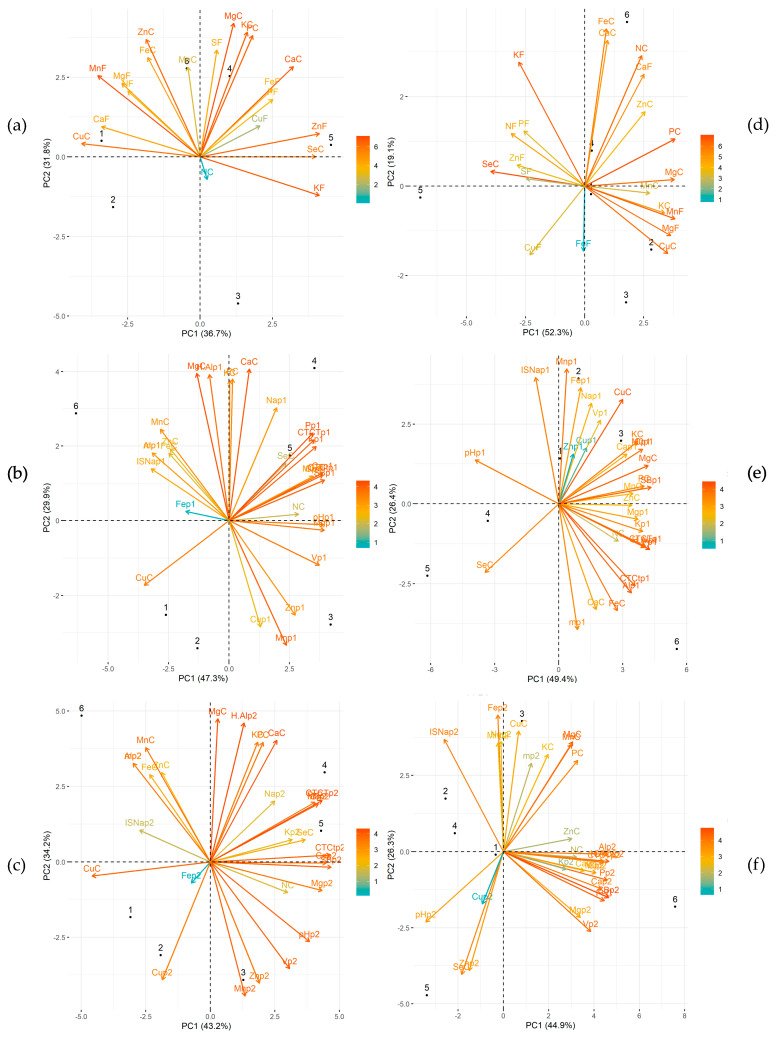
Principal component analysis (PCA) of chestnut and leaf minerals (**a**,**d**); chestnut and soil at depth p1 (0–20 cm) (**b**,**e**), and chestnut and soil at depth p2 (0–40 cm) (**c**,**f**) for *L. pisonis* (**a**–**c**) and *L. lanceolata* (**d**–**f**) trees. N, nitrogen; P, phosphorus; K, potassium; Ca, calcium; Mg, magnesium; Fe, iron; Zn, zinc; Cu, copper; Mn, manganese; S, sulfur; Se, selenium; pH, hydrogenionic potential in water; Al^+3^, exchangeable acidity; H + Al, potential acidity; C, carbon; Mo, organic matter; CTCt, effective cation exchange capacity; CTCT, cation exchange capacity; SB, sum of exchangeable bases; V, base saturation; ISNa, sodium saturation index for chestnut (C) and leaf (F).

**Figure 4 foods-12-04422-f004:**
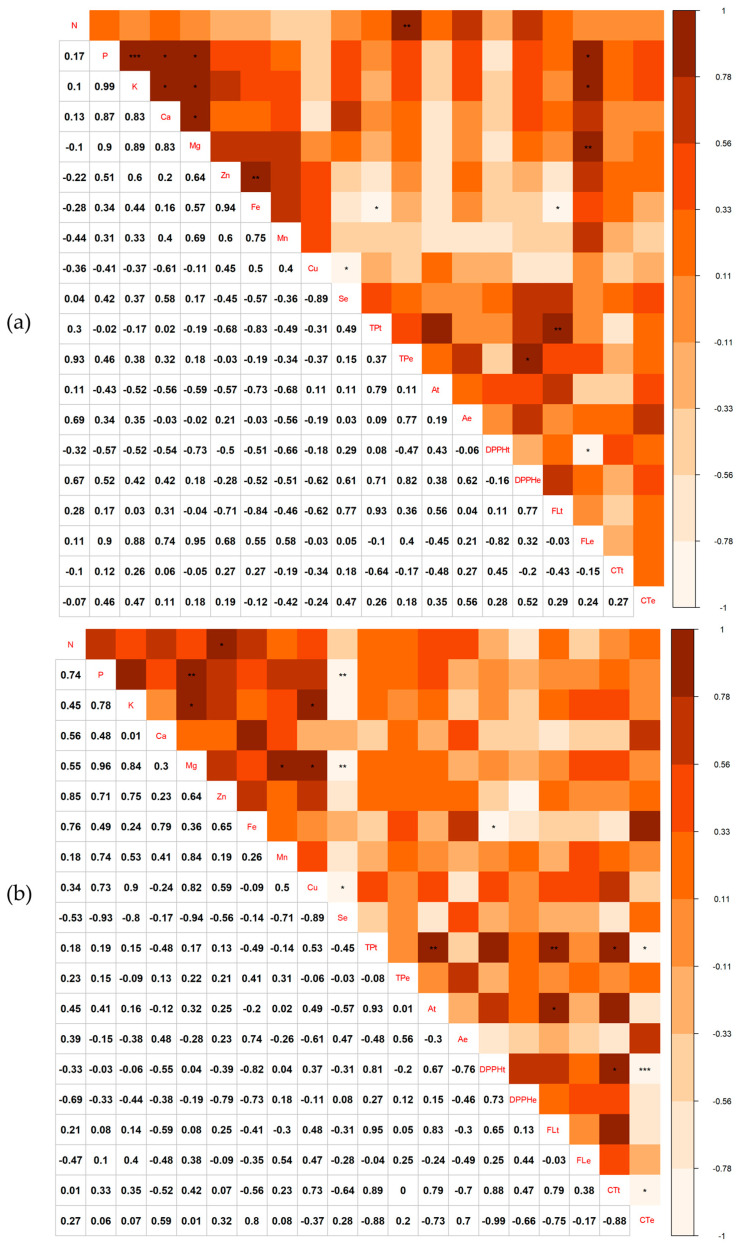
Pearson’s correlation between N, P, K, Ca, Mg, Fe, Zn, Cu, Mn, and Se content, total phenolics in the tegument (TPt) and endosperm (TPe), anthocyanins in the tegument (At) and endosperm (Ae), flavonols in the tegument (FLt) and endosperm (FLe), condensed tannins in the tegument (CTt) and endosperm (CTe), as well as 2,2-diphenyl-1-picrylhydrazyl in the tegument (DPPHt) and endosperm (DPPHe) of nuts of *L. pisonis* (**a**) and *L. lanceolata* (**b**). Significance levels were * 5%, ** 1%, and *** 0.1%.

**Figure 5 foods-12-04422-f005:**
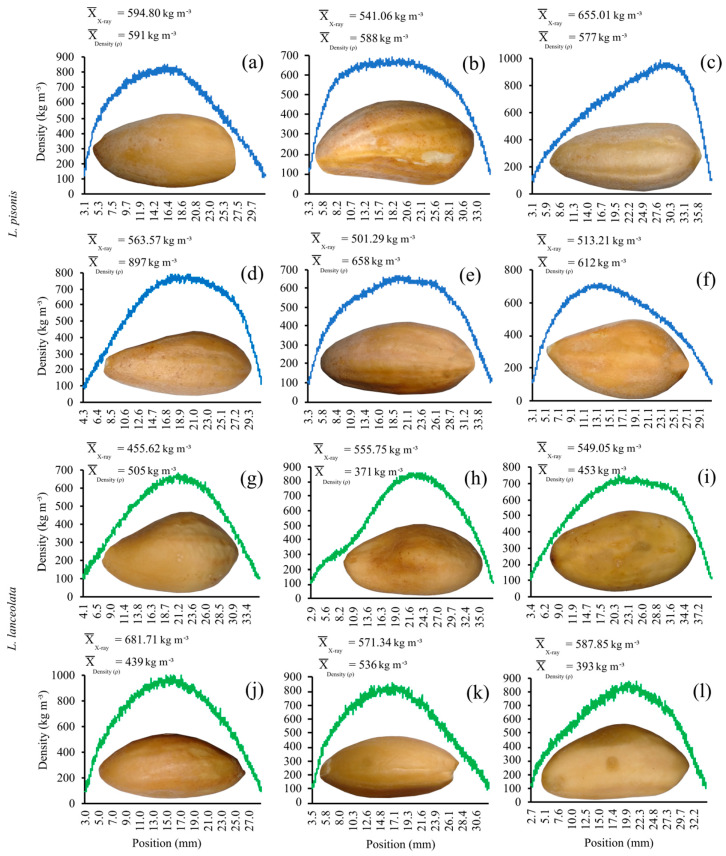
X-ray densitometry in nuts from *L. pisonis* trees (1, (**a**), 2 (**b**), 3 (**c**), 4 (**d**), 5 (**e**), and 6 (**f**)) and *L. lanceolata* (1 (**g**), 2 (**h**), 3 (**i**), 4 (**j**), 5 (**k**) and 6 (**l**)), starting in the apical region of the nut to the basal region (hilum).

**Figure 6 foods-12-04422-f006:**
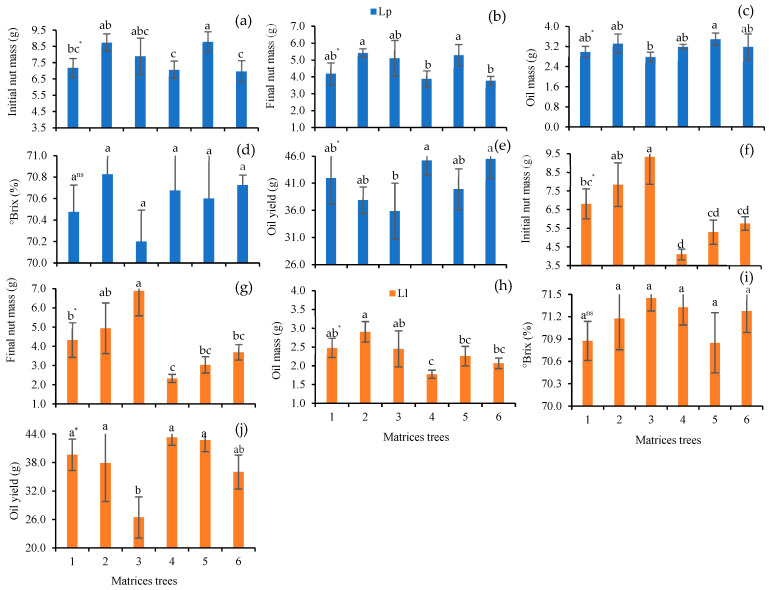
Characteristics of oil from fresh chestnuts of *L. pisonis* (Lp, **a**–**e**) and *L. lanceolata* (Ll, **f**–**j**) cold extracted using a hydraulic press, analyzing the of initial and final nut mass (**g**), oil mass (**g**), oil yield (**%**), and oil Brix (**%**). * Means followed by the same lowercase letter do not differ from each other by Tukey’s test (*p* < 0.05). Bar: standard deviation.

**Table 1 foods-12-04422-t001:** Fatty acid composition of *Lecythis pisonis* and *Lecythis lanceolata* oils.

***t_R_* (min)**	**Fatty acid**	** *L. pisonis* **					
		**1**	**2**	**3**	**4**	**5**	**6**
15.59	Myristic acid (C14:0)	11.01 ± 0.23	11.50 ± 0.34	12.10 ± 0.40	12.70 ± 0.03	12.00 ± 0.22	13.00 ± 0.08
19.88	Palmitic acid (C16:0)	0.24 ± 0.03	0.20 ± 0.01	0.60 ± 0.34	0.30 ± 0.01	0.30 ± 0.01	0.40 ± 0.01
20.41	Palmitoleic acid (C16:1)	0.10 ± 0.01	-	0.10 ± 0.01	0.10 ± 0.01	0.10 ± 0.01	-
22.19	Margaric acid (C17:0)	-	-	0.10 ± 0.01	-	-	-
22.71	Heptadec-10-enoic acid (17:1)	12.80 ± 0.08	6.50 ± 0.15	8.40 ± 0.06	11.70 ± 0.01	13.00 ± 0.11	7.60 ± 0.04
24.75	Stearic acid (C18:0)	36.50 ± 0.16	41.60 ± 0.25	52.40 ± 0.77	38.90 ± 0.05	40.20 ± 0.16	44.80 ± 0.02
25.22	Oleic acid (C18:1 n9c)	37.70 ± 0.19	38.50 ± 0.28	24.20 ± 0.07	34.40 ± 0.02	32.90 ± 0.07	32.10 ± 0.01
25.46	Vaccenic acid (C18:1 n11c)	1.30 ± 0.05	1.30 ± 0.01	1.70 ± 0.09	1.30 ± 0.02	1.10 ± 0.01	1.60 ± 0.03
26.39	Linoleic acid (C18:2 n6c)	0.10 ± 0.03	0.10 ± 0.01	0.10 ± 0.01	0.20 ± 0.01	0.10 ± 0.01	0.20 ± 0.02
27.85	Linolenic acid (C18:3)	0.20 ± 0.05	0.20 ± 0.01	0.20 ± 0.03	0.40 ± 0.03	0.30 ± 0.01	0.30 ± 0.01
	Total saturated fatty acids	47.70 ± 0.20	53.20 ± 0.10	65.30 ± 0.32	51.90 ± 0.25	52.40 ± 0.08	58.20 ± 0.07
	Total monounsaturated fatty acids	51.80 ± 0.15	46.40 ± 0.12	34.40 ± 0.28	47.50 ± 0.40	47.10 ± 0.07	41.30 ± 0.08
	Total polyunsaturated fatty acids	0.40 ± 0.03	0.30 ± 0.02	0.30 ± 0.03	0.50 ± 0.01	0.50 ± 0.01	0.50 ± 0.02
	S/U *	0.91 ± 0.04	1.14 ± 0.01	1.88 ± 0.04	1.08 ± 0.01	1.10 ± 0.01	1.39 ± 0.03
***t_R_* (min)**	**Fatty acid**	** *L. lanceolata* **					
		**1**	**2**	**3**	**4**	**5**	**6**
15.59	Myristic acid (C14:0)	9.30 ± 0.16	9.70 ± 0.23	10.30 ± 0.03	11.60 ± 0.15	9.80 ± 0.18	8.20 ± 0.15
19.88	Palmitic acid (C16:0)	0.10 ± 0.01	0.10 ± 0.03	0.10 ± 0.01	-	0.10 ± 0.01	0.10 ± 0.03
20.41	Palmitoleic acid (C16:1)	-	-	-	-	0.10 ± 0.01	-
22.19	Margaric acid (C17:0)	-	-	-	-	-	-
22.71	Heptadec-10-enoic acid (17:1)	3.70 ± 0.20	4.70 ± 0.05	5.60 ± 0.01	5.10 ± 0.03	6.10 ± 0.05	5.30 ± 0.07
24.75	Stearic acid (C18:0)	45.90 ± 0.17	45.10 ± 0.21	32.70 ± 0.05	31.80 ± 0.08	44.00 ± 0.11	50.50 ± 0.16
25.22	Oleic acid (C18:1 n9c)	39.40 ± 0.32	38.60 ± 0.01	49.50 ± 0.06	50.50 ± 0.10	38.50 ± 0.13	34.40 ± 0.02
25.46	Vaccenic acid (C18:1 n11c)	1.40 ± 0.01	1.40 ± 0.01	1.40 ± 0.05	1.00 ± 0.03	1.20 ± 0.02	1.00 ± 0.01
26.39	Linoleic acid (C18:2 n6c)	-	0.10 ± 0.03	0.10 ± 0.02	-	0.10 ± 0.01	0.10 ± 0.01
27.85	Linolenic acid (C18:3)	0.10 ± 0.01	0.20 ± 0.01	0.20 ± 0.02	-	0.20 ± 0.05	0.20 ± 0.03
	Total saturated fatty acids	55.40 ± 0.01	54.90 ± 0.01	43.10 ± 0.02	43.40 ± 0.10	53.96 ± 0.08	58.80 ± 0.04
	Total monounsaturated fatty acids	44.50 ± 0.05	44.80 ± 0.01	56.60 ± 0.03	56.60 ± 0.10	45.80 ± 0.09	40.80 ± 0.03
	Total polyunsaturated fatty acids	0.10 ± 0.01	0.30 ± 0.01	0.30 ± 0.01	0	0.30 ± 0.01	0.30 ± 0.01
	S/U *	1.24 ± 0.01	1.22 ± 0.02	0.76 ± 0.01	0.77 ± 0.07	1.17 ± 0.01	1.43 ± 0.05

* S/U: saturated to unsaturated ratio.

## Data Availability

Data is contained within the article and Appendix A.

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
