# Peer review of "Functional Fruit Trees from the Atlantic and Amazon Forests: Selection of Potential Chestnut Trees Rich in Antioxidants, Nutrients, and Fatty Acids"

_foods, 2023, doi:10.3390/foods12244422_

Round 1

Reviewer 1 Report

In this work, the authors investigate the Functional fruit trees of the Atlantic and Amazon Forest (Lecythidaceae) and discuss about the selection of potential chestnut trees rich in antioxidants, nutrients and fatty acids. The topic is interesting and the manuscript could be have potential from a scientific point of view. However, there are major flaws that require the authors’ attention.

1.      Line 43: Please give a few examples regarding the beneficial effects of functional foods.

2.      Introduction: I believe it is important that some more general information about antioxidants, nutrients and fatty acids (and their health impact) should be added in the Introduction section. After all, these elements are highlighted also in the title.

3.      Lines 74-78: The aim paragraph should be edited in order to present the main aims of the study in full clarity. At its current state, it gives the impression of guessing and downgrades the authors’ effort.

4.      2.1: In my opinion, the latitude and longitude coordinates should be added for each case (in-text).

5.      Subsections 2.5 to 2.8: Appropriate references should be added

6.      Figure 2 is quite interesting but values are impossible to read. Can you make it bigger or break it into more figures? This is very bad resolution. The same applies for figures 3 and 4.

7.      Lines 275-290: Due to the state of Figure 2 I can not be sure, but wouldn’t all this information be better shown on a Table? The authors should consider it.

8.      In general, the whole Results section is chaotic. Try to create appropriate subsections in order to make it easier for the reader.  (e.g. minerals, fatty acids, etc)

9.      The discussion section is well organized and informative. I would prefer to see more literature regarding the elements of interest (about their health effects), if available. Ideally though, I believe some literature about the potential technological use of chestnuts in functional products (eg in ceral bars or cookies, etc) should be presented.

10.   Conclusions should be improved. Start with a summary of results (not too technical), and finalize with an outlook. More precise and comprehensive conclusions can be provided for readers. At its current state this section is rather poor, but the scientific picture given in the manuscript is more complex.

Minor editing required.

Author Response

  1. Line 43: Please give a few examples regarding the beneficial effects of functional foods.

The suggestion was answered.

  1. Introduction: I believe it is important that some more general information about antioxidants, nutrients and fatty acids (and their health impact) should be added in the Introduction section. After all, these elements are highlighted also in the title.

The suggestion was answered.

  1. Lines 74-78: The aim paragraph should be edited in order to present the main aims of the study in full clarity. At its current state, it gives the impression of guessing and downgrades the authors’ effort.

The suggestion was answered.

  1. 2.1: In my opinion, the latitude and longitude coordinates should be added for each case (in-text).

The suggestion was answered.

  1. Subsections 2.5 to 2.8: Appropriate references should be added

The suggestion was answered.

  1. Figure 2 is quite interesting but values are impossible to read. Can you make it bigger or break it into more figures? This is very bad resolution. The same applies for figures 3 and 4.

The suggestion was answered.

  1. Lines 275-290: Due to the state of Figure 2 I can not be sure, but wouldn’t all this information be better shown on a Table? The authors should consider it.

The suggestion was answered.

  1. In general, the whole Results section is chaotic. Try to create appropriate subsections in order to make it easier for the reader. (e.g. minerals, fatty acids, etc)

The suggestion was answered.

  1. The discussion section is well organized and informative. I would prefer to see more literature regarding the elements of interest (about their health effects), if available. Ideally though, I believe some literature about the potential technological use of chestnuts in functional products (eg in ceral bars or cookies, etc) should be presented.

The suggestion was answered.

  1. Conclusions should be improved. Start with a summary of results (not too technical), and finalize with an outlook. More precise and comprehensive conclusions can be provided for readers. At its current state this section is rather poor, but the scientific picture given in the manuscript is more complex.

The suggestion was answered.

Reviewer 2 Report

The study entitled “Functional fruit trees of the Atlantic and Amazon Forest (Lecythidaceae): selection of potential chestnut trees rich in antioxidants, nutrients, and fatty acids” focuses on the Brazilian tropical forests as an abundant source of unconventional fruit trees, particularly Lecythis pisonis and Lecythis lanceolata, which produce functional nuts. The authors aim to identify potential functional nut donor trees by evaluating the nutritional and antioxidant composition of the nuts, as well as the fatty acid profile of the oil. While the paper successfully presents valuable findings regarding the characteristics of these nuts and their potential as functional foods, there are certain limitations and areas of concern that need to be addressed. Please below find my remarks:

·       The introduction section should be rewritten in order to improve the clarity and flow of the text. The other sections are well written.

·       Please provide references for the methods used to determine all the elements. Are they standard methods, as for example the determination of N with the Dumas method that is ISO certified?

·       Figure 2: contains too much information. Please provide the figures separately (perhaps as supplementary material) to ensure the validity of the results.

·       Figure 3: please provide a higher resolution.

·       Lines 315-317 please provide the values in the text.

·       Figure 4: Please provide a better resolution.

·       Figure 5: Please consider to be shown as supplementary material.

·       Figure 8: Please elaborate further on what you are showing in Figure 8. Is every bar an individual measurement or an average. Why error bars are missing?

·       Table 1: No standard deviations, no statistical analysis and comparison among the trees are available.

·       Overall, the authors are presenting many results that cannot be evaluated in that form. Please consider moving some of the results to the supplementary section, or convert certain figures to tables.

The text should be checked for grammatical and linguistic errors.

Author Response

The study entitled “Functional fruit trees of the Atlantic and Amazon Forest (Lecythidaceae): selection of potential chestnut trees rich in antioxidants, nutrients, and fatty acids” focuses on the Brazilian tropical forests as an abundant source of unconventional fruit trees, particularly Lecythis pisonis and Lecythis lanceolata, which produce functional nuts. The authors aim to identify potential functional nut donor trees by evaluating the nutritional and antioxidant composition of the nuts, as well as the fatty acid profile of the oil. While the paper successfully presents valuable findings regarding the characteristics of these nuts and their potential as functional foods, there are certain limitations and areas of concern that need to be addressed. Please below find my remarks:

  • The introduction section should be rewritten in order to improve the clarity and flow of the text. The other sections are well written.

The suggestion was answered.

  • Please provide references for the methods used to determine all the elements. Are they standard methods, as for example the determination of N with the Dumas method that is ISO certified?

The suggestion was answered.

  • Figure 2: contains too much information. Please provide the figures separately (perhaps as supplementary material) to ensure the validity of the results.

The suggestion was answered.

  • Figure 3: please provide a higher resolution.

The suggestion was answered.

  • Lines 315-317 please provide the values in the text.

The suggestion was answered.

  • Figure 4: Please provide a better resolution.

The suggestion was answered.

  • Figure 5: Please consider to be shown as supplementary material.

The suggestion was answered.

  • Figure 8: Please elaborate further on what you are showing in Figure 8. Is every bar an individual measurement or an average. Why error bars are missing?

The suggestion was answered.

  • Table 1: No standard deviations, no statistical analysis and comparison among the trees are available.

The suggestion was answered.

  • Overall, the authors are presenting many results that cannot be evaluated in that form. Please consider moving some of the results to the supplementary section, or convert certain figures to tables.

The suggestion was answered.

Reviewer 3 Report

Dear Authors,

The manuscript entitled "Functional fruit trees of the Atlantic and Amazon Forest (Lecythidaceae): selection of potential chestnut trees rich in antioxidants, nutrients and fatty acids" and signed by Caroline Palacio de Araujo et al. presents the results of research on the nutritional value and pro-health properties of nuts from two Lecythis varieties (L. pisonis and L. lanceolata). Despite its average originality, the topic and findings are important to the scientific community and humanity. In terms of editing, the work was prepared with great care.  Some judgements and indications for improvement of the manuscript are presented below.

Abstract

The aims are clearly presented. The main results and conclusions are also indicated. Only the repetition of what is stated in the manuscript (lines 24-25 and 27) is objectionable.

1.       Introduction

The authors describe in detail selenium (lines 64-73) - its content in the plant and its use in the human body. Why don't they mention anything about the other macro- and micronutrients they mean in nuts?

I also suggest adding a broader paragraph about antioxidant properties, which are also determined for nuts. Why is it so important?

The authors hypothesize that the tested nuts can be classified as functional foods. Please specify what exactly this statement means.

In line 56, the world chestnut extraction value is given. In which year?

2.       Materials and methods

line 104 - ratio nitric acid: perchloric acid was a volume ratio?

line 108 - please enter the concentration of sulfuric acid

lines 109-110 Please specify precisely how the P content was determined after reading the absorbance value on the spectrophotometer

line 123 - To determine N content, 0.1 g sample was digested... - what sample? Concentrated extract?

line 158 - concentration of HCl is given in N and other reagents in M - please harmonize

line 169 - FL is not in equation 1

lines 172 and 175 - please add numbers to these and other equations

all work - there should be no space between the temperature value and the degree symbol

lines 191-192- The treatments consisted of a blank (1 mL filtrate + 2 mL ethanol) and three triplicates (1 mL filtrate + 2 mL 150 μM DPPH). - what was treated? - please explain

lines 222-223 - split mode split - is this correct?

3.       Results

The obtained results are described and discussed carefully. On the other hand, the figures and charts that are included raise huge reservations. In such small graphs, the reader is not able to see anything. I suggest giving up some of them in favor of tables or reorganizing them. Maybe put some of the results in supplementary materials? I understand that it requires a lot of work, but in this form they do not fulfill their function.

4.       Discussion

line 484 - where does the absorption and transport of nutrients to leaves and seeds come from?

lines 491-498 - how do the tested nuts compare to the presented values?

line 499 - what pH?

line 503 - what diseases?

Best regards,

Reviewer

Author Response

Abstract

The aims are clearly presented. The main results and conclusions are also indicated. Only the repetition of what is stated in the manuscript (lines 24-25 and 27) is objectionable.

The suggestion was answered.

  1. Introduction

The authors describe in detail selenium (lines 64-73) - its content in the plant and its use in the human body. Why don't they mention anything about the other macro- and micronutrients they mean in nuts?

The suggestion was answered.

I also suggest adding a broader paragraph about antioxidant properties, which are also determined for nuts. Why is it so important?

The suggestion was answered.

The authors hypothesize that the tested nuts can be classified as functional foods. Please specify what exactly this statement means.

The suggestion was answered.

In line 56, the world chestnut extraction value is given. In which year?

The suggestion was answered.

  1. Materials and methods

line 104 - ratio nitric acid: perchloric acid was a volume ratio?

The suggestion was answered.

line 108 - please enter the concentration of sulfuric acid.

The suggestion was answered.

lines 109-110 Please specify precisely how the P content was determined after reading the absorbance value on the spectrophotometer.

The suggestion was answered.

line 123 - To determine N content, 0.1 g sample was digested... - what sample? Concentrated extract?

The suggestion was answered.

line 158 - concentration of HCl is given in N and other reagents in M - please harmonize.

The suggestion was answered.

line 169 - FL is not in equation 1.

The suggestion was answered.

lines 172 and 175 - please add numbers to these and other equations.

The suggestion was answered.

all work - there should be no space between the temperature value and the degree Symbol.

The suggestion was answered.

lines 191-192- The treatments consisted of a blank (1 mL filtrate + 2 mL ethanol) and three triplicates (1 mL filtrate + 2 mL 150 μM DPPH). - what was treated? - please explain.

The suggestion was answered.

lines 222-223 - split mode split - is this correct?

The suggestion was answered.

  1. Results

The obtained results are described and discussed carefully. On the other hand, the figures and charts that are included raise huge reservations. In such small graphs, the reader is not able to see anything. I suggest giving up some of them in favor of tables or reorganizing them. Maybe put some of the results in supplementary materials? I understand that it requires a lot of work, but in this form they do not fulfill their function.

The suggestion was answered.

  1. Discussion

line 484 - where does the absorption and transport of nutrients to leaves and seeds come from?

Plants need at least 14 essential minerals for optimal functioning and these are absorbed by their roots. The root system of dicotyledonous plants is formed by the primary root, in which the lateral roots also develop (Osmont et al., 2007). The absorption of these nutrients can occur at the tip of the root, as well as in root hairs that act by increasing the contact surface with the soil. Nutrients are transported radially to the vascular system, with the xylem responsible for transporting water and minerals to the aerial part of the plant, including the leaves. The phloem is responsible for transporting carbohydrates, hormones and other organic compounds produced in the leaves to other parts of the plants, such as seeds (Taiz et al., 2015; Robe; Barberon, 2023).

ROBE, K.; BARBERON, M. Nutrient carriers at the heart of plant nutrition and sensing. Current Opinion in Plant Biology, v. 74, p. 102376, 2023.

OSMONT, K.S.; SIBOUT, R.; HARDTKE, C.S. Hidden branches: developments in root system architecture. Annual Review Plant Biology, v. 58, p. 93-113, 2007.

Taiz, L.; Zeiger, E.; Moller, I.M.; Murphy, A. (2015) Plant Physiology and Development. 6th Edition, Sinauer Associates, Sunderland, CT.

lines 491-498 - how do the tested nuts compare to the presented values?

All mentioned are nuts that are already inserted and accepted in the food market.

line 499 - what pH?

The suggestion was answered.

line 503 - what diseases?

The suggestion was answered.

Round 2

Reviewer 1 Report

The authors have addressed my comments.

Minor editing required.

Reviewer 3 Report

Dear Authors,
thank you for the detailed answers to my doubts. In my opinion, the presented version of the publication does not raise much controversy and can be published in the  Foods journal.
Here are some comments that were noticed while reading the manuscript:
line 56 - repetition of becoming a source
Fig 2 - please explain the ns symbol in the Nitrogen content chart
line 506 - no dot between degrees and the word Brix

Best regards,

Reviewer
